# The Role of Endoscopic Ultrasonography in the Diagnosis and Staging of Pancreatic Cancer

**DOI:** 10.3390/cancers14061373

**Published:** 2022-03-08

**Authors:** Ali Zakaria, Bayan Al-Share, Jason B. Klapman, Aamir Dam

**Affiliations:** 1Department of Gastroenterology-Advanced Endoscopy, H. Lee Moffitt Cancer Center, Tampa, FL 33612, USA; jason.klapman@moffitt.org (J.B.K.); aamir.dam@moffitt.org (A.D.); 2Department of Hematology and Oncology, Karmanos Cancer Center, Wayne State University, Detroit, MI 48201, USA; alshareb@karmanos.org

**Keywords:** pancreatic cancer, endoscopic ultrasonography, loco-regional staging

## Abstract

**Simple Summary:**

Pancreatic cancer ranks as the seventh leading cause of cancer-related death worldwide. The overall prognosis of pancreatic cancer is poor even in patients with resectable disease. Early detection and accurate diagnosis and staging of pancreatic cancer are essential for developing a treatment strategy, as surgical resection can provide the only potential cure for this disease. Endoscopic ultrasonography is essential in the diagnosis and loco-regional staging of the disease.

**Abstract:**

Pancreatic cancer is the fourth leading cause of cancer-related death and the second gastrointestinal cancer-related death in the United States. Early detection and accurate diagnosis and staging of pancreatic cancer are paramount in guiding treatment plans, as surgical resection can provide the only potential cure for this disease. The overall prognosis of pancreatic cancer is poor even in patients with resectable disease. The 5-year survival after surgical resection is ~10% in node-positive disease compared to ~30% in node-negative disease. The advancement of imaging studies and the multidisciplinary approach involving radiologists, gastroenterologists, advanced endoscopists, medical, radiation, and surgical oncologists have a major impact on the management of pancreatic cancer. Endoscopic ultrasonography is essential in the diagnosis by obtaining tissue (FNA or FNB) and in the loco-regional staging of the disease. The advancement in EUS techniques has made this modality a critical adjunct in the management process of pancreatic cancer. In this review article, we provide an overall description of the role of endoscopic ultrasonography in the diagnosis and staging of pancreatic cancer.

## 1. Background

Pancreatic cancer ranks as the seventh leading cause of cancer-related death worldwide, and the fourth among other cancers in the United States in both men and women [1,2,3]. Pancreatic ductal adenocarcinoma (PDAC) accounts for the majority (85%) of these tumors [1,4]. In 2022, approximately 62,210 new diagnoses of exocrine pancreatic cancer are predicted in the United States [1]. It is projected that within the next two decades, pancreatic cancer incidence rates will continue to increase and surpass colorectal cancer to be the second leading cause of cancer-related deaths in the United States [5].

The most challenging aspect of diagnosis of PDAC is the late presentation. Most patients present at an advanced stage with overall 5-year survival of approximately 10%. Candidates for curative surgery at diagnosis are only 10–15% of cases [5,6]. The most important prognostic factor is the tumor stage at diagnosis, supporting the role of early detection and role of screening in pancreatic cancer. Recent data from the National Cancer Institute revealed a significant increase in the proportion of people diagnosed with stage IA pancreatic cancer from 2004 to 2016 and improvement in 5-year overall survival for this stage from 44.7% in 2004 to 83.7% in 2012 [7]. The current guidelines of the United States Preventive Services Task Force and other societies recommend against screening except in patients considered high-risk because of family history or diagnosis of an inherited genetic syndrome [8].

The advancement of imaging studies and involving a multidisciplinary team approach involving radiologists, gastroenterologists, medical, radiation, and surgical oncologist have a major impact on the management of pancreatic cancer. Endoscopic ultrasonography (EUS) is essential in the diagnosis (fine needle aspiration or biopsy (FNA or FNB)) and loco-regional staging of the disease. In addition, advancement in EUS techniques has made this modality a vital adjunct in the management process of pancreatic cancer.

## 2. Introduction

Invasive PDAC arises from precancerous precursor lesions referred to as Pancreatic Intraepithelial Neoplasia (PanIN) that form by metaplasia and proliferation of ductal epithelium. These lesions gradually progress to higher degrees of dysplasia by acquiring mutations and genetic changes and can ultimately develop into invasive PDAC [9,10,11,12]. PanIN-1 and PanIN-2 are considered low grade while PanIN-3 is considered a high-grade lesion with higher malignant potential [13,14]. The majority of PanINs develop in the small intralobular ducts where they cause lobulocentric acinar atrophy, obstructing the small duct secretions and causing local inflammation of the surrounding pancreatic tissue [15]. Despite the malignant potential, only a small proportion of low-grade PanIN lesions transform to cancer [9]. Incidence increases with age and higher-grade lesions are more common in individuals with a strong family history of pancreatic cancer [16].

When PDAC develops, initial presentation varies by the site of the tumor. Sixty to 70% of tumors develop in the head of the pancreas, and they present with signs of biliary obstruction and exocrine pancreatic insufficiency [17,18,19,20]. In contrast to tumors in the body and tail, which occur in 20–25% of cases, individuals present with more nonspecific symptoms and typically at a later stage [17].

The standard diagnostic test is the multi-detector computerized tomography (MDCT) scan with pancreatic protocol to guide the initial evaluation of a suspected pancreatic mass and staging of the disease [21,22,23]. Although it is usually an adequate diagnostic tool, it’s not highly sensitive for lesions smaller than 2 cm, which makes it a poor screening tool [24]. Endoscopic Ultrasound directly visualizes the mass and is used to obtain tissue via FNA or FNB to confirm the diagnosis. In addition, EUS can characterize the T and N staging of the lesion, and aid in the decision of upfront resectability versus a neoadjuvant approach in patients with pancreatic cancer [25]. Positron emission tomography (PET) scan can help identify patients with metastatic disease. Due to its low specificity, a PET scan is not helpful to identify tumor size, invasion, and nodal staging [26,27]. The T staging is better assessed by the MDCT [28].

Multiple guidelines exist for the management of pancreatic neoplasia once a suspicious pancreatic mass is found, however; there has been no consensus on the optimal role of EUS in the diagnosis and staging of pancreatic cancer.

## 3. Risk Factors

The risk of pancreatic cancer increases with several environmental or modifiable factors and other nonmodifiable genetic factors. Among the modifiable factors, cigarette smoking appears to have the strongest association [29]. Smokers have a 1.5 relative risk of developing pancreatic cancer compared to nonsmokers, and the risk increases with an increased number of cigarettes used [3,30,31,32]. Alcohol consumption appears to have a small increase in risk that is limited to heavy drinkers, although this association is thought to be confounded by cigarette smoking in these patients [33,34]. Furthermore, increased body mass index (BMI), low physical activity, and impaired fasting glucose are also associated with increased risk of pancreatic cancer [29,35,36,37,38]. The association between diabetes and pancreatic cancer is well studied. While some data suggest that diabetes is a consequence of undiagnosed pancreatic cancer, most of the evidence suggests that hyperglycemia and insulin resistance precede the diagnosis of pancreatic cancer [39,40,41,42,43]. In addition, the degree of hyperglycemia has been associated with worse pathologic features and survival in pancreatic cancer [44,45]. Diabetic patients who have pancreatic cancer may have worse pathologic features and survival [46]. One retrospective review suggested that screening CT scans at the time of diabetes onset can reveal resectable pancreatic cancer [47]. However, only a minority of patients with diabetes age 50 and older will be diagnosed with pancreatic cancer.

Chronic pancreatitis is a well-known risk factor of pancreatic cancer, however; less than 5% of patients with chronic pancreatitis develop pancreatic cancer which questions the role of screening in these patients [48,49,50,51]. Neoplastic pancreatic cysts carry an increased risk of malignancy. The most common cyst type, intraductal papillary mucinous neoplasms (IPMNs) are defined as growth within the pancreatic ducts and characterized by the production of mucin. IPMNs can arise either in the main or branch portion of the pancreatic duct. Surgery is generally recommended for the main duct subtype or the branch duct subtype with high-risk features (obstructive jaundice, enhancing mural nodule ≥ 5 mm nodule) as these lesions carry up to a 62% risk of malignancy [52,53,54,55]. The branch duct subtype has an overall lower risk of malignant potential and surveillance is generally recommended for lesions without worrisome or high-risk features. Other pancreatic cystic neoplasms are called mucinous cystic neoplasms (MCNs) carry a <15% risk of malignancy, although resection is still recommended in all fit patients who have MCNs [55,56,57]. Hereditary pancreatic cancer includes patients with either an inherited genetic syndrome with a recognized germline mutation or a familial pattern of pancreatic cancer in which a family has at least two affected first-degree relatives with no known genetic predisposition syndromes. Only 5–10% of patients with pancreatic cancer have a hereditary risk [55,58]. Table 1 summarizes the cancer predisposition syndromes and the relative risk of incidence of pancreatic cancer.

Individuals with familial risk in the absence of cancer predisposition syndrome have a relative risk of 4.5 for the development of pancreatic cancer compared to the general population if one first-degree relative (FDR) is affected, and 6.4-fold risk if 2 FDRs are affected. The risk is as high as 32-fold if three or more FDRs are affected [73].

## 4. Screening

The current guidelines recommend against screening for pancreatic cancer in the general population because of its overall low prevalence rate [7]. Different societies recommend screening in the high-risk population, with subtle differences. In 2019, The international Cancer of the Pancreas Screening (CAPS) consortium updated their recommendations on screening in the high-risk population. The goal of screening is to identify the high-grade dysplastic lesions and the T1N0M0 cancers [74].

Candidates for screening include all patients with Peutz Jeghers syndrome, carriers of a germline *CDKN2A* mutation, carriers of a germline *BRCA2*, *BRCA1*, *PALB2*, *ATM*, *MLH1*, *MSH2*, or *MSH6* gene mutation with at least one affected first-degree blood relative. In addition, screening is recommended for individuals who have at least one first-degree relative with pancreatic cancer who in turn also has a first-degree relative with pancreatic cancer, and those with three or more relatives affected (familial pancreatic cancer kindred) [74]. Age of starting screening varies among risk factors: Individuals with Peutz Jeghers syndrome or *CDKN2A* mutations are recommended to start screening at age of 40. Carriers of *BRCA2*, *ATM*, *PALB2*, *BRCA1*, *MLH1*, or *MSH2* start screening at age of 45 or 50 or 10 years younger than the youngest affected FDR. And people with known familial pancreatic cancer with no germline mutations start at age of 50 or 55 or 10 years younger than the youngest affected blood relative [74]. Both MRI/MRCP and EUS were recommended at baseline, in addition to fasting blood glucose and/or HbA1c. While there were no specific recommendations during follow-up, alternating MRI/MRCP and EUS were recommended. EUS-FNA was recommended for solid lesions, cystic lesions with concerning features, and asymptomatic main pancreatic duct stricture of unknown etiology. Intervals of screening are 12 months, however; if there were concerning abnormalities with no indication for immediate surgery, intervals should be shortened to 3 or 6 months [74].

The most recent American College of Gastroenterology (ACG) guidelines agreed with the CAPS expert recommendations [75]. The 2020 American Gastroenterological Association (AGA) guidelines recommended initial age of screening to start at age 50 or 10 years younger than the initial age of familial onset, age of 40 years for carriers of *CKDN2A* and *PRSS1* mutations with hereditary pancreatitis, and age 35 years in the setting of Peutz Jeghers syndrome [76]. EUS and MRI roles in surveillance are considered complementary and not interchangeable. This is based on two prospective studies that enrolled patients with high risk for PDAC for screening [77,78]. EUS detected more solid lesions than MRI/MRCP, while MRI/MRCP was more sensitive to cystic lesions of any size [77,78].

Recently, circulating tumor DNA (ctDNA) has been suggested as a helpful biomarker for screening and post-treatment surveillance in different cancer types, including PDAC [79] although experts have not adopted this approach in clinical practice until further data is available [74].

## 5. Diagnosis

The diagnosis of pancreatic cancer should be suspected in patients who present with weight loss, anorexia, abdominal pain, jaundice, dark urine, and/or steatorrhea. The signs of PDAC on examination include jaundice, hepatomegaly, cachexia, abdominal mass, nontender palpable gallbladder (Courvoisier’s sign), and/or ascites. The initial presentation of PDAC varies according to the tumor location (head/uncinate vs body tail) [18]. The recent onset of diabetes mellitus has been recognized as a presenting finding in patients with PDAC and should trigger further evaluation [80]. Occasionally, a solid pancreatic mass can be incidentally found on abdominal images performed for unrelated indications. Other rare presentations include migratory superficial thrombophlebitis (Trousseau’s syndrome), and paraneoplastic skin manifestations [81].

The initial diagnostic workup of patients with suspected pancreatic cancer includes serological workup (amylase, lipase, liver function test, and CA 19-9), and abdominal imaging studies. In general, single imaging modalities do not seem to be adequate for the diagnosis of preoperative staging of pancreatic cancer. Helical CT scan seems to be the most accurate in evaluating the extent of the primary tumor, locoregional extension, vascular invasion, distant metastases, tumor TNM stage, and tumor resectability. EUS complements CT scan and is accurate in assessing tumor size, degree of vascular involvement, and lymph node involvement [82]. Th sensitivity and positive predictive value for the detection of pancreatic cancer differ among existing imaging modalities. Transabdominal ultrasound is commonly used as the initial imaging modality for evaluation of a symptomatic patient, however; diagnosis of pancreatic cancer using this modality is highly operator dependent and detection of pancreatic lesions also varies with body habitus and location of the tumor. It has a variable sensitivity that ranges between 68% and 95% and a variable specificity that ranges between 50% and 100% [83,84,85]. CT scan with dual-phase pancreas protocol has a sensitivity of 76–92% and specificity of 67% for the diagnosis of pancreatic cancer, respectively [86,87,88]. Historically, a CT scan has shown better accuracy than angiography in demonstrating tumor involvement of major peripancreatic vessels. An accurate diagnosis was established in 91% of cases with 8% false positive and 1% false-negative rates [89]. Despite the improvement in sensitivity and the predictive value of determining resectability using MDCT, the challenge with using CT scan as a single modality is the poor detection of lymph node metastasis, microscopic, localized tumor invasion, and peritoneal metastasis [90,91]. CT scan has a sensitivity for detection of lymph node metastasis of 19–37% and specificity of 60–92% [89,92,93,94,95,96]. Sensitivity did not improve using a short axis diameter of 5 m and 10 mm as criteria for lymph node involvement [97,98].

The role of MRI has been compared to other modalities in diagnosis and determining tumor resectability. A meta-analysis of sixty-eight studies showed a lower sensitivity of MRI compared to CT scan in diagnosis but comparable sensitivity in determining resectability. Another meta-analysis of 5399 patients from 52 studies showed equivalent sensitivity of both modalities for determining a diagnosis. MRI has a sensitivity of 84–93% and specificity of 82–89% in diagnosing pancreatic cancers [86,87,88,99]. A possible advantage of diffusion-weighted MRI over CT scan is the detection of small liver lesions not detected on CT, as it can detect liver metastasis in 1.5–2.3% of patients with negative CT scans on presurgical evaluation [100]. MRI is also useful in the detection of isoattenuating lesions, a hypertrophied pancreatic head, and the detection of focal fatty parenchymal infiltration [101]. Gadoxedic acid-enhanced MRI has 92–94% sensitivity in detecting liver metastasis compared to 74–76% sensitivity of CT scan, and it is better at distinguishing hepatic micro-abscess from metastasis [102,103]. The National Comprehensive Cancer Network (NCCN) guidelines recommend MRI use as an adjunct to CT scan in individual cases when further characterization of indeterminate liver lesions on CT scan is needed, evaluation of the suspected tumor is not visible on CT scan, or in patients with iodine allergy [104].

^18^FDG PET/CT scan is used in high-risk patients after pancreatic protocol CT scan or MRI and is recommended by NCCN guidelines to detect extrapancreatic tumor metastasis. In some cases, it can change lymph node staging [105]. High-risk patients include those who have symptoms concerning systemic disease and patients with markedly elevated CA19.9, borderline resectable disease, large primary, and large regional lymph nodes [104,106].

Endoscopic ultrasonography is currently considered the most sensitive imaging study for the detection of pancreatic lesions. Multiple studies revealed that conventional EUS using radial or linear echoendoscopes and its related techniques such as contrast-enhanced EUS and EUS elastography has a major role in the diagnosis and staging of pancreatic cancer. A systematic review of 9 studies (*n* = 678) revealed that the sensitivity of EUS and CT scan was 91–100% and 53–91% in the detection of pancreatic adenocarcinoma, respectively [107]. The higher sensitivity of EUS compared to CT scan was also described in 19 comparative studies (*n* = 895) [108,109]. The EUS sensitivity to detect pancreatic lesions < 2 cm was 94.4% compared to 50% of CT scans. The high-resolution images and the superior sensitivity of the EUS translated into better detection of smaller pancreatic lesions (1–2 cm), indeterminate lesions, and lesions not identified on CT scan or other imaging modalities [110,111,112]. In addition, it has been shown that the absence of a focal mass lesion on EUS reliably excluded pancreatic cancer with a negative predictive value of 100% [113].

The endosonographic appearance of pancreatic adenocarcinoma is typically described as a heterogeneous hypoechoic solid mass with irregular borders; however, this appearance is not specific for adenocarcinoma Figure 1.

The new advancement in EUS technology with the addition of ultrasound contrast enhancer agent in association with low mechanical index techniques improved the ability of endosonographer to differentiate between benign and malignant pancreatic lesions and to perform advanced therapeutic procedures. The ability of dynamic contrast-enhanced harmonic endoscopic ultrasonography (CE-EUS) to differentiate between pancreatic lesions was based on the fact that more than 90% of PDAC cases are hypovascular in nature. PDAC lesions are visualized as hypoechoic hypoenhancing lesions on CE-EUS, while pseudotumoral lesions due to chronic pancreatitis, autoimmune pancreatitis, and neuroendocrine pancreatic tumors are either isoechoic isoenhancing or hyperechoic hyperenhancing. In a prospective multicenter study (*n* = 100), the performance of CH-EUS was compared with that of EUS-FNA for the diagnosis of pancreatic adenocarcinoma. The accuracy, sensitivity, specificity, positive predictive value, and negative predictive value of CE-EUS in diagnosing PDA was 95%, 96%, 94%, 97%, and 91%, respectively. CE-EUS was also able to correctly diagnose five cases with false-negative EUS-FNA [114]. In another retrospective study (*n* = 394) that evaluated the etiology of small solid pancreatic lesions (≤15 mm) to optimize their management, CE-EUS allowed differential diagnosis of PDAC and non-PDAC in 86% of cases [115]. In a meta-analysis (12 studies, *n* = 1139) evaluated the accuracy of CE-EUS for diagnosing PDAC in patient with pancreatic mass, the pooled sensitivity was 94% [95% CI: 91–95%], specificity was 89% [95% CI: 85–92%], positive likelihood ratio was 8.09 [95% CI: 4.47–14.64], and the negative likelihood ratio was 0.08 [95% CI: 0.06–0.10] [116]. A recent systematic review and meta-analysis (16 studies, *n* = 1325) evaluated the diagnostic performance of CH-EUS for the differentiation of pancreatic masses. The CE-EUS had a high accuracy in differentiating between malignant and benign pancreatic tumors with a pooled sensitivity of 93% [95 % CI: 91–94%], specificity of 84% [95% CI: 80–87%], the positive likelihood ratio of 5.58 [95% CI: 3.90–7.97], the negative likelihood ratio of 0.09 [95% CI: 0.07–0.11] [117].

The role of CE-EUS in improving the diagnostic yield of EUS guided tissue acquisition was presumed based on the better visualization of the targeted area in the lesions by avoiding the inside necrosis and the vessels of fibrosis. In a small prospective study, the diagnostic accuracy of pancreatic lesions using CE-EUS-FNA guidance was 86.5% compared to 78.4% when conventional EUS-FNA was used (*p* = 0.35). There were two false-negative EUS-FNA cases correctly identified by CH-EUS [118]. In a prospective RCT, the accuracy and sensitivity of CE-EUS-FNA in the diagnosis of solid pancreatic lesions compared with that of conventional EUS-FNA were not statistically significant. However, a sufficient sample was obtained with just one pass in 60% of patients using CE-EUS-FNA compared with only 25% with conventional EUS FNA [119]. In a more recent RCT (*n* = 240) there was no statistically significant difference between the diagnostic sensitivity of CE-EUS-FNA 85.3% and conventional EUS-FNA 88.3% (*p* = 0.564). Most patients in the CE-EUS-FNA arm and all patients in the conventional EUS-FNA arm received a diagnosis within 3 needle passes [120]. Overall, CE-EUS can be used to enhance the diagnostic accuracy especially if this technology is used selectively for small, indeterminate solid pancreatic lesions.

Despite the presence of multi societal guidelines for the management of pancreatic neoplasia once a suspicious pancreatic mass is found, there has been no consensus on the optimal approach for diagnosis and staging. Multiple clinical algorithms exist with differences in regard to the role of EUS in the workup of pancreatic cancer.

The NCCN latest guidelines stated that the gold standard initial imaging for evaluation of suspicious pancreatic cancer is the multi-detector helical CT angiography with thin submillimetric (preferably < 3 mm) sections, performed with a dual phase pancreatic protocol (pancreatic and portal venous phase of contrast enhancement) using oral and IV contrast. The role of EUS in the staging of pancreatic cancer is felt to be complementary to pancreatic protocol CT scan, hence EUS is not recommended as a routine staging tool and should not be used to assess vascular involvement [104].

On the other hand, the American Society for Gastrointestinal Endoscopy (ASGE) in their latest statement regarding the role of endoscopy in the evaluation and management of a patient with solid pancreatic neoplasia recommended the imaging evaluation of pancreatic neoplasia should include both EUS and multidetector pancreas protocol CT scan. The use of EUS was further recommended particularly when CT scan detection of resectability is equivocal [121]. The Canadian Society of Endoscopic Ultrasound proposed an algorithm for diagnosis and staging of pancreatic cancer and their recommendation was to strongly consider EUS and EUS guided biopsy to achieve the most precise diagnosis, maximize the chance to identify distant metastasis and provide tissue diagnosis if the neoadjuvant approach is considered [122]. Also, the Japanese Pancreas Society in their latest clinical practice guidelines for pancreatic cancer recommended EUS as a diagnostic test in subjects with suspected pancreatic cancer because it is more sensitive than other imaging modalities [123].

## 6. EUS-Guided Tissue Acquisition

### 6.1. Indication

The pathologic diagnosis of pancreatic cancer is not required in surgical candidates with diagnostic images reflecting resectable disease. Furthermore, a nondiagnostic pathological sample in a patient with high clinical suspicion for pancreatic cancer should not delay surgery in patients with resectable disease [124,125].

However, several reports revealed a rate of 5–10% of patients were misdiagnosed for pancreatic cancer and underwent surgical resection for benign diseases such as chronic pancreatitis and lymphoplasmacytic sclerosing pancreatitis, or other malignant etiologies that requires chemotherapy rather than surgery such as lymphoma [126,127]. In addition, there is an emerging role of neoadjuvant therapy in the management of pancreatic cancer (see below), especially in patients with lesions that might become potentially resectable. These observations can lead to the need for tissue diagnosis in all patients with pancreatic lesions.

The current guidelines recommend tissue diagnosis as a mandatory step in patients with borderline resectable or locally advanced disease before neoadjuvant chemotherapy, and in patients with unresectable or metastatic disease before initiating chemotherapy. These two groups of patients account for up to 85% of patients with pancreatic cancer [125,128].

### 6.2. Method for Tissue Acquisition

Tissue samples can be performed using a fine needle aspiration through an endoscopic (EUS-guided) or percutaneous (US or CT- guided) approach. The EUS-guided approach is preferred by multiple international societies due to better diagnostic accuracy, lower risk of infection, superior overall safety, lower potential risk of tumor seeding, and the additional staging information during EUS evaluation [125,129,130]. EUS-FNA is the preferred method to obtain tissue for diagnosis of pancreatic cancer even when other methods revealed nondiagnostic samples. EUS-FNA has a high sensitivity of 80–95% and a specificity of 96–100% for the evaluation of solid pancreatic masses [131,132,133,134,135]. A large meta-analysis of 20 studies (*n* = 2761) showed a sensitivity of 90.8% and specificity of 96.5% [136].

EUS-FNA can be performed using multiple available needles with different sizes ranging from 19–25 gauge (Table 2). The choice of needle size can be dependent on the site of the pancreatic mass. For example, the use of the 19-gauge needle can be challenging if the mass is in the pancreatic head/uncinate process area due to the decreased maneuverability and stiff nature of this needle, in such cases, 25 or 22-gauge needles can be of an advantage. The diagnostic yield of different needle sizes has been evaluated in multiple studies. One study revealed no difference in diagnostic yield for pancreatic and peripancreatic lesions between the 19-gauge vs 22-gauge needles. Four meta-analyses compared 25-gauge to 22-gauge needles and only one meta-analysis (*n* = 1292, 22-gauge = 799 and 25-gauge = 565) revealed that 25-gauge needle has a statistically significant higher sensitivity for diagnosing malignancy compared to 22-gauge needle (pooled sensitivity 0.93 vs 0.85; *p* = 0.0003) [137]. The other three studies showed a trend to higher sensitivity for the 25-gauge needle but did not reach statistical significance [138,139,140]. However, both needle sizes are recommended for FNA of pancreatic masses.

EUS-FNB needles have been designed to obtain core biopsies samples with preservation of tissue architecture and increased cellularity. The NCCN and ASGE guidelines recommend that EUS-FNB should be considered if EUS-FNA is nondiagnostic and a histologic diagnosis is required for further ancillary tests such as molecular and genomic studies [104,114,118]. Three newer generation FNB needles are available with varying bevel designs of the distal tip of the needle to facilitate the acquisition of core samples (Table 2). A recent meta-analysis (*n* = 1365) revealed superior diagnostic accuracy without compromising safety when comparing FNB to FNA. The study showed higher diagnostic accuracy (87% vs. 81%, *p =* 0.005), better cytopathologic accuracy (89% vs. 82%, *p* = 0.04), decreased number of mean passes required (1.6 vs. 2.3, *p <* 0.0001), and comparable safety profile [140]. Another randomized crossover trial (*n* = 140) comparing the diagnostic yield of FNB to FNA for pancreatic and non-pancreatic lesions revealed an overall superior diagnostic yield and sample adequacy for all solid mass lesions obtained by FNB; however, the diagnostic yield difference was not statistically significant for pancreatic masses (91.7% vs. 78.4%, *p =* 0.19). The success of crossover to the FNB arm was 96.5% (*p =* 0.0003). This study also showed the cost-effectiveness of FNB compared to FNA [141].

The development of miniature biopsy forceps allowed performing EUS guided through the needle forceps biopsy (EUS-TTNFB). This entails using a 19-gauge FNA needle to puncture the lesion and once the location of the needle is confirmed, the biopsy forceps are advanced through the needle into the lesion and bites can be obtained under EUS guidance. The feasibility and use of the microforceps have been studied in the evaluation of pancreatic cystic lesions, however; the role of the microforceps in solid pancreatic lesions is still unclear. A pilot study showed the technical feasibility and safety of EUS-TTNFB in tissue acquisition from solid lesions including pancreatic masses. The tissue acquisition rate of EUS-TTNFB was up to 100% per session (3 passes). The diagnostic accuracy of EUS-TTNFB and EUS-FNA for malignancy with a single pass was 83%, and with a single session (3 passes) was 94% [142,143]. Further studies will be needed before adopting this method of tissue sampling.

### 6.3. Techniques during Tissue Acquisition

After the FNA or FNB needle is introduced into the pancreatic lesion, certain maneuvers and techniques have been described to enhance tissue acquisition. Different no suction and suction techniques have been evaluated. The stylet slow pull technique entails slowly and continuously withdrawing the stylet from the needle to create minimal negative pressure while the endosonographer is moving the needle to-and-fro within the target lesion. The suction technique entails using a 10- to 20-mL syringe with negative pressure as dry suction or wet suction (the needle will be prefilled with saline to enhance negative pressure transmission to the tip of the needle). A randomized control study (*n* = 121) compared stylet slow pull (*n* = 61) to suction techniques (*n* = 60) revealed comparable technical success, diagnostic sensitivity, first pass diagnosis rate, acquisition of core tissue, and the median number of passes to diagnosis between both techniques [144]. The use of standard negative pressure suction technique in malignant pancreatic masses resulted in higher sensitivity and diagnostic yield compared to no suction technique [145,146]. A randomized controlled study revealed significantly better cellularity and sample adequacy in a cell block for wet suction techniques compared to conventional FNA suction [147]. Another study compared high negative pressure (using a 50 mL syringe) to standard negative pressure suction and revealed similar diagnostic accuracy; however, high negative pressure provided the sample with more tissue, and the technique required statistically fewer passes to reach a diagnosis [148].

The puncture and actuation technique can also influence the sample adequacy and diagnostic yield. Different puncture techniques and the number of actuations has been evaluated. The standard technique entails passing the needle within the target lesion to-and-fro in the same trajectory of the initial puncture. The fanning technique on the other hand entails passing the needle in four different trajectories to obtain tissue from different areas of the lesion. A randomized controlled study revealed a better diagnostic yield when increased actuations (15 vs. 10) combined with no suction technique was used. There was no statistical difference in diagnostic yield of malignancy between 10, 15, or 20 actuations combined with the suction technique, although there were more bloody samples observed when the number of actuations increased [149]. Another randomized trial (*n* = 54) compared fanning (*n* = 26) to standard (*n* = 28) technique revealed lower number of required passes to establish the diagnosis and higher percentage of achieving diagnosis on first pass (85.7% vs. 57.7%, *p =* 0.02) [150].

### 6.4. Onsite Pathologic Evaluation Methods

The presence of rapid onsite evaluation (ROSE) by a cytopathologist was proposed to improve the diagnostic yield of EUS tissue sampling from patients with solid pancreatic lesions. The added value of ROSE is still controversial. Moreover, the implication of cost and availability of trained cytopathologists are other limiting factors for this approach. Older studies in 2003 and 2005 reported improved diagnostic yield of EUS-FNA when the on-site cytopathologic evaluation was implemented [151,152]. Another meta-analysis (34 studies, *n* = 3644) revealed that ROSE increases the diagnostic accuracy of EUS-FNA for pancreatic adenocarcinoma [131,153]. On the contrary, more recent randomized controlled studies that evaluated the EUS-FNA with and without ROSE revealed no significant difference in diagnostic yield or adequacy between the two groups. Furthermore, these studies demonstrated that a lower number of passes in the ROSE group did not indicate a lower complication rate or lower cost [154,155]. Two recent meta-analyses also revealed that ROSE does not improve the diagnostic accuracy of EUS-FNA for pancreatic masses [132,155]. A noninferiority randomized controlled study conducted at 14 centers in 8 countries compared EUS-FNB of solid pancreatic lesions with and without ROSE (*n* = 771, with ROSE = 385 vs. without ROSE = 386) revealed comparable diagnostic accuracy, safety, and sample quality in both groups. A higher tissue core rate and shorter mean sampling procedure time was obtained by EUS-FNB without ROSE [156]. Another multicenter noninferiority randomized controlled study compared EUS-FNB alone to EUS-FNA with ROSE (*n* = 235, EUS-FNB alone = 115, EUS-FNA with ROSE = 120) revealed that the diagnostic accuracy of EUS-FNB lone is non-inferior to EUS-FNA with ROSE. It was also associated with fewer passes, shorter procedure time, and comparable cost [157].

The introduction of new generation FNB needles provided macroscopic onsite evaluation (MOSE) as a more practical alternative to ROSE. This approach entails that the endosonographer confirms the presence of a core tissue from the EUS-FNB sample obtained after each pass. A retrospective study revealed that MOSE with a 22-gauge Franseen tip FNB needle demonstrated higher diagnostic adequacy and accuracy of >90% without ROSE [158]. A randomized controlled trial comparing EUS-FNB with MOSE vs EUS-FNA with ROSE is ongoing with an estimated completion date is September 2023 (ClinicalTrials.gov Identifier: NCT03766659). The histopathologic evaluation of samples obtained by EUS-guided FNA or FNB can be challenging, the use of an FNB needle can provide more tissue for satisfactory diagnosis and further ancillary testing (Figure 2A–D). Other promising onsite pathological evaluation methods under investigation include telecytopathology and artificial intelligence (AI) using an automated visual inspection system [159,160].

## 7. Staging

Staging and evaluation of the extent of the disease are essential in determining the treatment plan and identifying candidates for curative surgery. The standard staging system in pancreatic cancer is the American Joint Committee on Cancer (AJCC) 8th edition that follows the Tumor/Node/Metastasis system (TNM). The most recent edition revised the T staging to size-based rather than descriptive definitions, describing the size as the best surrogate of tumor biology after resection based on two large multi-institutional series that indicted that tumor size reflects survival [161,162,163]. It also changed recommendations on the T4 category that is now considered unresectable [161].

Defining the resectability of locoregional disease was addressed by different groups. The Americas Hepatopancreaticobiliary Association/Society for Surgery of the Alimentary Tract/Society of Surgical Oncology (AHPBA/SSO/SSAT), the MD Anderson group, and the NCCN guidelines defined unresectable disease based on radiographic criteria with the differences between group definitions mainly in defining superior mesenteric vein (SMV) involvement [104,164,165,166]. In 2013, the intergroup radiographic definition was set for the Alliance A0201101 pilot study to eliminate subjective terminology and have better criteria set for the clinical trials [167]. This defined the borderline resectable disease as a localized disease with one or more of the following [167]:An interface between the primary tumor and superior mesenteric vein portal vein (SMV-PV) measuring 180° or greater of the circumference of the vein wall, *and/or*Short segment occlusion of the SMV-PV with normal vein above and below the level of obstruction that is amenable to resection and venous reconstruction, *and/or*Short segment interface (of any degree) between tumor and hepatic artery with normal artery proximal and distal to the interface that is amenable to resection and arterial reconstruction, *and/or*An interface between the tumor and superior mesenteric artery (SMA) or celiac trunk measuring less than 180° of the circumference of the artery wall.

The definition of locally advanced unresectable disease includes [104]:Solid tumor contact with SMA or celiac artery (CA) > 180°.Pancreatic body/tail contact with the CA and aortic involvement.Unreconstructible SMV/PV due to tumor involvement or occlusion.

The role of EUS in the staging of PDAC has been reinforced by multiple studies [168]. In meta-analyses, the pooled sensitivity and specificity of EUS in the detection of tumor vascular invasion were 66–86% and 89–94%, respectively [169,170]. The sensitivity of EUS varies based on the blood vessel involved. It is superior to a CT scan for assessing portal vein involvement; however, it is less accurate for evaluating the relationship of the mass to the superior mesenteric artery, superior mesenteric vein, and celiac artery Figure 3 [171,172,173]. In a study of 62 patients that underwent R0 resection, EUS alone identified borderline resectable PDAC in 29% of patients, CT scan alone identified 23% and both modalities identified 48% of patients. EUS identified 11 patients who required vein reconstruction that CT scan did not identify [174]. EUS is also sensitive for nodal staging Figure 4. A meta-analysis of 16 studies (*n* = 512) revealed a pooled sensitivity and specificity of 69% and 81% respectively for detection of a lymph node. In 8 of these studies (*n* = 281), EUS revealed higher sensitivity than CT scan [169]. Distant metastasis is better detected by CT scan and MRI; however, EUS has a better detection rate of small left hepatic metastatic lesions and small pockets of ascites otherwise not detected on CT scan [175,176]. EUS detection of ascites can alter management and prognosis. One retrospective study (*n* = 42) revealed 41% of the patients were found to have liver metastasis by EUS-FNA that were not previously found on other imaging modalities [177].

## 8. Role of Neoadjuvant Therapy

The current gold standard treatment for potentially resectable PDAC is surgical resection of the tumor followed by adjuvant chemotherapy. However, this approach has been recently challenged by the fact that 40–60% of patients who underwent surgical resection never received the planned adjuvant chemotherapy. This can be due to postsurgical deconditioning or complications [178]. These observations along with the fact that systemic chemotherapy is a cornerstone in the management of PDAC, new studies have been advocating for a neoadjuvant approach, where chemotherapy and radiotherapy are provided prior to surgery in potentially resectable disease [179].

The goal of neoadjuvant therapy in PDAC is to improve patients’ overall survival by improving tumor resectability with negative margins [180,181]. Neoadjuvant chemotherapy and/or chemoradiation are established in the current guidelines for borderline resectable disease [181,182]. The role of radiation therapy is continuously evolving. Stereotactic body radiation therapy (SBRT) is a modality being used in patients with borderline and locally advanced pancreatic cancer. For patients receiving SBRT, fiducials (metallic markers) are placed inside or nearby the tumor to serve as a reference point during radiation therapy Figure 5. Historically, fiducial markers were placed surgically or, percutaneously, under CT or ultrasound guidance. Most recently, EUS has been utilized as the preferred approach for fiducial placement with high technical success and low risk for adverse events [183,184].

Surgery after neoadjuvant treatment seems to be safe and this approach might increase the likelihood of patients receiving systemic therapy with lower incidence of delays, lower incidence of pancreatic fistulas, and having an R0 resection [185,186,187,188,189,190]. The role of neoadjuvant therapy is emerging in resectable disease, and a few trials have shown promising results of safety and local control [191,192,193,194,195]. Multiple other trials are currently ongoing. Histologic confirmation is necessary per guidelines prior to neoadjuvant therapy, and EUS-FNA/FNB is the preferred approach [104].

## 9. Conclusions and Future Direction

Pancreatic cancer is a leading cause of cancer-related death worldwide. Early detection and accurate diagnosis and staging of pancreatic cancer are of paramount importance for planning treatment options. Despite the lack of consensus on the optimal role of EUS in the diagnosis and staging of pancreatic cancer among society guidelines, EUS is essential in the diagnosis by tissue acquisition (FNA or FNB) and in the loco-regional staging of the disease. The diagnostic adequacy and accuracy can be enhanced by the choice of the needle type, tissue acquisition method, and technique.

The advancement in EUS technology led to tremendous improvement in the diagnostic accuracy of PDAC. The new era of personalized medicine will have the most influence on the advancement of the EUS field for both diagnostic and therapeutic purposes. The development of a novel fusion imaging system that uses electromagnetic sensors to combine live EUS images with preprocedural CT scan images can improve navigation and target difficult pancreatic lesions. Another promising technology is the implementation of machine learning algorithms to provide an automatic computer-aided diagnosis in real-time during EUS evaluation of solid pancreatic lesions. This technology can significantly improve the efficiency of EUS evaluation of pancreatic masses. The field of EUS guided tissue acquisition became of paramount importance, especially with more need to perform predictive molecular markers or cell culture with chemosensitivity testing to guide individualized therapies. This was the major factor that led to the improvement and invention of available FNA and FNB needles. The onsite pathological evaluation methods are a hot area of evaluation with new technologies being under investigation, including telecytopathology and artificial intelligence (AI) using an automated visual inspection system.

## Figures and Tables

**Figure 1 cancers-14-01373-f001:**
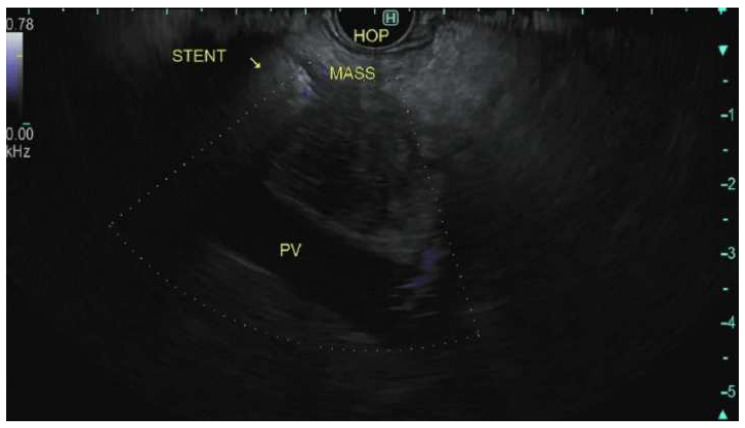
Endoscopic Ultrasonographic images using linear echoendoscope reveal a heterogeneous hypoechoic solid mass with irregular borders in the head of the pancreas.

**Figure 2 cancers-14-01373-f002:**
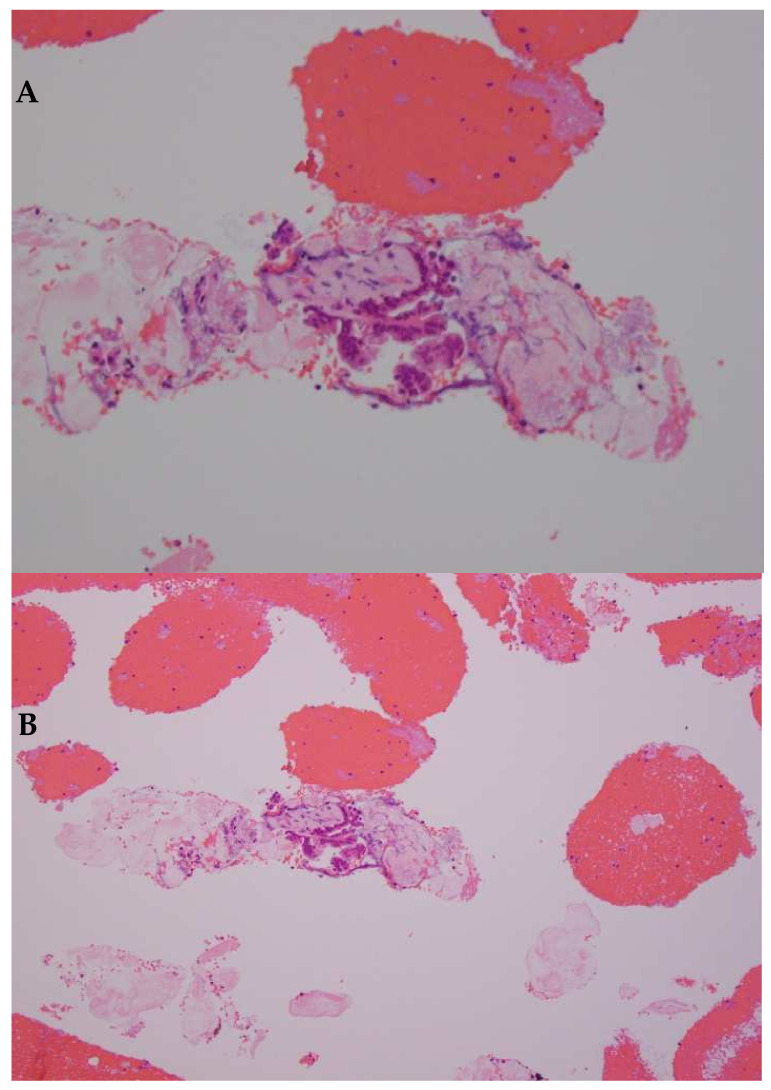
Cell blocks from FNA (**A**,**B**) vs FNB (**B**,**C**) needles. (**A**) FNA cell block with single fragment with desmoplastic stroma and cluster of adenocarcinoma cells surrounded by blood (hematoxylin and eosin, 10×). (**B**) The sample contains a limited number of malignant cells, sufficient for a diagnosis of malignancy, but insufficient for ancillary testing (hematoxylin and eosin, 20×) (**C**) Sample collected using FNB needle, intact core fragments fill the field, in comparison to the FNA sample (hematoxylin and eosin, 10×) (**D**) The increased number of representative cells renders this adequate for ancillary testing (hematoxylin and eosin, 20×) (Images courtesy of Dr. Barbara A. Centeno, Department of Pathology, Moffitt Cancer Center, Tampa, FL, USA).

**Figure 3 cancers-14-01373-f003:**
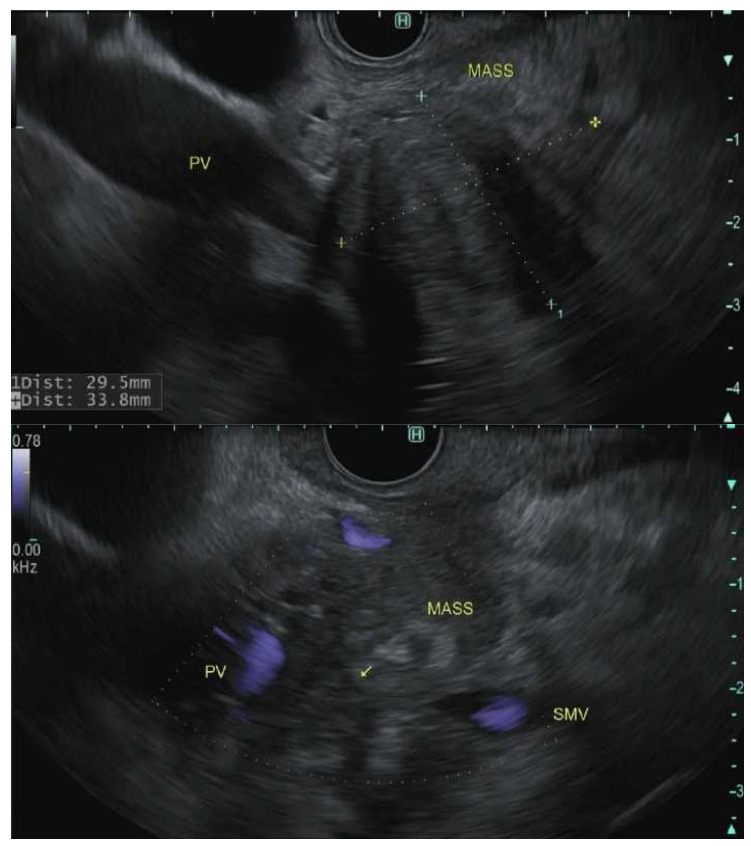
Endoscopic Ultrasonographic images using linear echoendoscope reveal pancreatic mass with the invasion of the portosplenic confluence to different degrees in all three illustrations.

**Figure 4 cancers-14-01373-f004:**
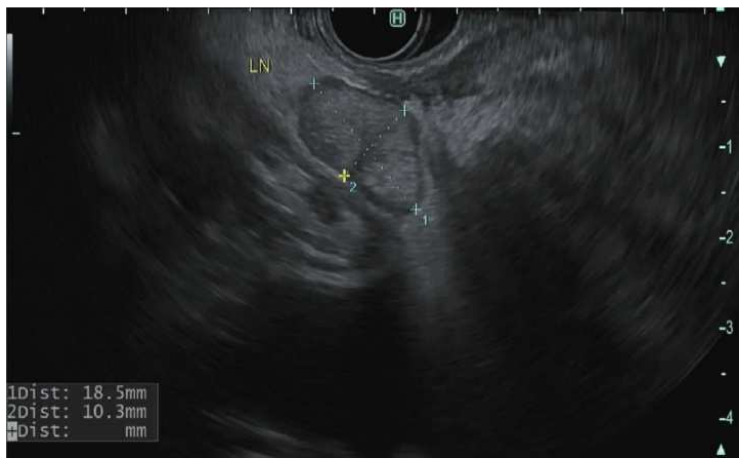
Endoscopic Ultrasonographic images using linear echoendoscope reveal peripancreatic lymph nodes.

**Figure 5 cancers-14-01373-f005:**
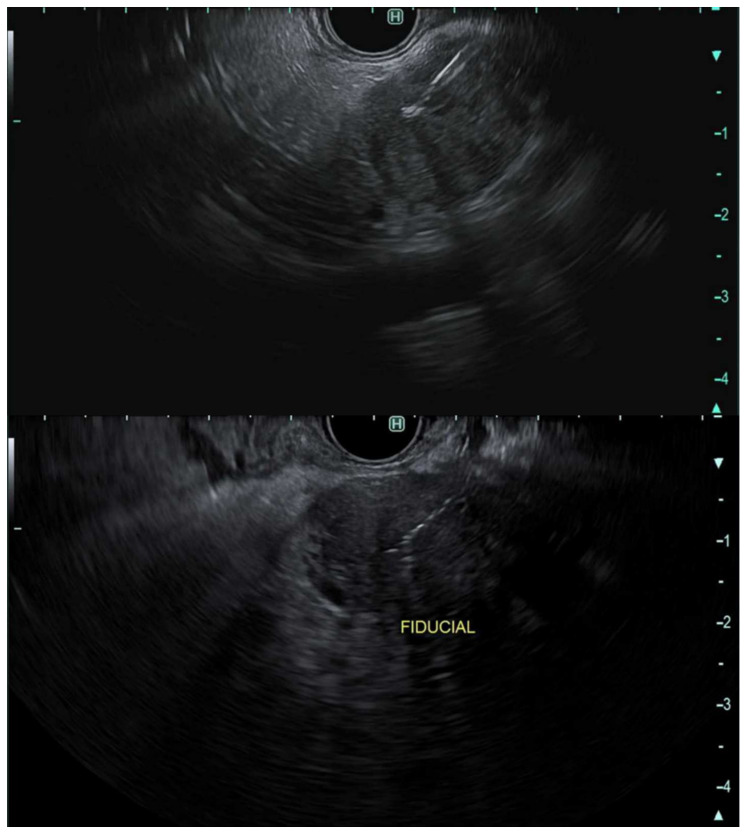
Endoscopic Ultrasonographic images using linear echoendoscope reveal pancreatic mass with a fiducial marker placed using EUS guidance.

**Table 1 cancers-14-01373-t001:** Cancer predisposition syndromes and relative risk of pancreatic cancer.

Cancer Predisposition Syndrome	Gene Involved	The Relative Risk of Pancreatic Cancer
Peutz Jeghers syndrome	*STK11*	132 [59,60]
Lynch syndrome	Mismatch repair genes (*MLH1, MSH2, MSH6*)	8.6 [61]
HBOC *	*BRCA1*	2.26 [62]
HBOC *	*BRCA2*	3.5–6.6 [63,64]
HBOC *	*PALB2*	Unknown [64,65,66]
FAMMM **	*CKDN2A*	13–22 [67]
Ataxia telangiectasia	*ATM*	Unknown [68]
Li Fraumeni	*TP53*	7 [69]
Hereditary pancreatitis	*PRSS1, SPINKI*	50–82 [70,71,72]

* HBOC: Hereditary Breast and Ovarian Cancer, ** FAMMM: familial atypical melanoma mole syndrome.

**Table 2 cancers-14-01373-t002:** EUS fine-needle aspiration (FNA) and fine needle biopsy (FNB) needles are available in the USA.

EUS-FNA Needles			
Needle	Manufacturer	Caliber	Photo
Expect^TM^	Boston Scientific	19G, 22G, 25G	a 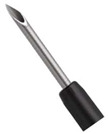 a 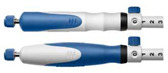
Expect^TM^ Flex		19G	
Expect^TM^ SlimLine (SL)		19G, 22G, 25G	
Expect^TM^ SlimLine Flex		19G	
ClearView^TM^ Round	ConMed	19G, 22G, 25G	b 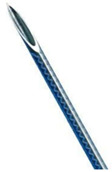
ClearView^TM^ Round w/sheath stabilizer		22G, 25G	
ClearView^TM^ Extended bevel		22G	
ClearView^TM^ Extended bevel w/sheath stabilizer		22G	
Beacon^TM^ FNA needle	Medtronic	19G, 22G, 25G	c 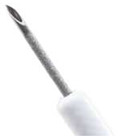
Beacon^TM^ EUS delivery system with FNA needle		19G, 22G, 25G	
EchoTip® Ultra	Cook Medical	19G, 22G, 25G	d 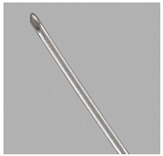
EchoTip® Ultra Coil sheath		22G	
EchoTip® Ultra HD access		19G	d 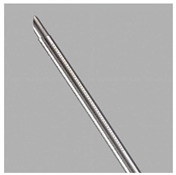
EZ Shot 2	Olympus	19G, 22G, 25G	
EZ Shot 2 with side hole		22G	e 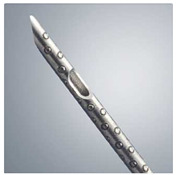
EZ Shot 3 Plus with and without side hole		19G, 22G	e 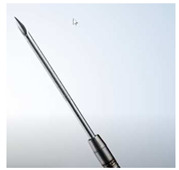
**EUS-FNB Needles**			
**Needle**	**Manufacturer**	**Caliber**	**Photo**
Acquire^TM^	Boston Scientific	22G, 25G	a 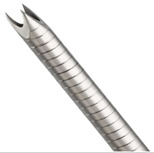
SharkCore^TM^ FNB needle	Medtronic	19G, 22G, 25G	c 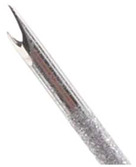
SharkCore^TM^ FNB biopsy system with needle		22G, 25G	
Beacon^TM^ EUS delivery system with SharkCore^TM^ LG FNB needle		19G	
EchoTip ProCore® Ultra HD	Cook Medical	19G, 20G, 22G, 25G	d 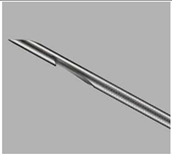

a Reprinted Courtesy of Boston Scientific Corporation. b Reprinted Courtesy of ConMed Corporation. c Reprinted Courtesy of Medtronic Corporation. d Reprinted Courtesy of Cook Medical Corporation. e Reprinted Courtesy of Olympus Corporation.

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
