# Peer review of "The Role of Endoscopic Ultrasonography in the Diagnosis and Staging of Pancreatic Cancer"

_cancers, 2022, doi:10.3390/cancers14061373_

Round 1

Reviewer 1 Report

Dear Authors,

   Happy Lunar New Year 2022! Thanks so much for manuscript submission to MDPI Journal of Cancers. In sum, the comprehensive study is complete and fair, the literal writing is also fine. However, from the perspective of a professional research journal, there are quite a few aspects need to be addressed. My recommendation of this initial draft is between major revision and minor revision.

   This paper presents an overview of endoscopic ultrasonography in the diagnosis and staging of pancreatic cancer. The authors specified some reported results (i.e., data on 5-year survival), the advancement of imaging studies and the multidisciplinary approach.  Besides, the formatting issues of the entire context need to be reshaped with respect to the MDPI template on Cancers. The suggested edits towards major and minor aspects (about the problematic issues) are summarized as below (may not limited to these).

   Major problematic issues suggested for edits in your updated version:

   a) It is fine to be preserved with current length of between 180-200 words, some keynote concluding remarks and major quantitative results should be addressed. Despite it is just a review article, I suggest an alternative plan to present the abstract by "Objective: ... Methods: .... Approach ... Conclusion ..."

   b) Introduction: need to provide more specific details on the review of prior work, and the organization of this paper should be stated. Besides, some journals may prefer to specify a summary of keynote contributions. Please consider a major rewrite and apply the corresponding changes.

   c) I think the subsections of "Risk factors", "Screening" and "Diagnosis", may be combined to the Section 2 of "Related Work" or "Related Studies". The number on section of "EUS-Guided Tissue Acquisition" is also missing.

   d) Figures and Tables: While the four figures are well-shaped, the size and position should be calibrated. Also, the title of "Figure X", should start from the beginning of the next line, and preserve the required space behind each figure (along with detailed descriptions). Regarding the tables, I have two suggestions: a) don't let a table cross over multiple pages (i.e., *HBOC at Line 103 ...), if one table is too long, use the template for compact format of the linespacing and specify "Table continued" at the beginning of the next page. If any more tabulated results are to be summarized, please supplement any.

   e) Discussions: since it is a review article (survey work on journal), some discussions are necessary. If the subsequent part on "Role of neoadjuvant therapy" belongs to this section, qualitative analysis along with commenting on the limitations of study, are suggested to be added.

   f) Conclusions: the current section is a bit too generic. It lacks summary of keynote work or your contributions, missed the future studies (not a single sentence at the last). Hence, I suggest the authors preparing a major rewrite on the last paragraph, along with adding some more specific details on the research challenges and a few orientations of prospective research tasks.

   g) References: the cited journal papers should use abbreviated format for the title of each journal; and the cited conferene papers should contain accurate information on the time and location of each proceedings; please supplement each of them. Meanwhile, I think the coverage of References took up too many pages, please arrange them with respect to the MDPI template. Besides, if adding 5-6 state-of-the-art methods published in Years 2020-2022 (better MDPI journal publications), it should be better. 

   Minor issues to be updated in your revised version:

   a) Stop hyphenating words which may transit to the next line. Hyphens to connect broken words which crossed over two adjacent lines, should be calibrated. You may adjust the MDPI template (MS word) for this issue. If you are using latex, check the options to fix that problem.
   b) Use the MDPI template to provide the whole part of context, neither of the current font type nor the size is correct. You should use size 10, Palatino Linotype. Attribute the titles with the Sections and subsections, respectively. 
   c) In some context, literal quality of English writing needs further improvement, along with grammatical checks and proofreading. 

   d) Apply the italic format of some notations such as "n = ..." at Line 280; for other numerical scores, please apply the uniform valid digits on accuracy.

   Once again, thank you, best of luck to your further edits. We are looking forward to seeing your updated manuscript coming into minor revision and later acceptance!

Stay safe,

With warm regards,

Reviewer 2 Report

This paper describes EUS modality for diagnosing pancreatic cancer. It is comprehensively well documented. What is expected from the title is detailed information of EUS including diagnostic effectiveness for pancreatic cancer. General information for diagnosing pancreatic cancer such as a risk factor is certainly important, but the author should describe EUS as a diagnostic method more in detail including contrast enhanced EUS and usefulness for suspecting carcinoma in site should be also described.

Round 2

Reviewer 2 Report

This paper is revised properly. It describes the diagnosis of pancreatic cancer comprehensively including EUS, while the method of EUS-FNA is explained in detail. The title should be considered or separate the descruption about EUS and EUS-FNA.

Author Response

Dear reviewer,

Thank you for your feedback. We decided on the title based on the request/invitation from the journal to provide a review article regarding role of EUS in pancreatic cancer. As you mentioned the manuscript describe the major role of EUS in pancreatic cancer diagnosis and staging. As you know, EUS-FNA is a cornerstone step in the process of pancreatic cancer diagnosis, we specifically detailed the literature in that regard as its an evolving process. We kept the title more general to describe all aspects of diagnosis.

Thanks